# Synthesis of Spin-Labeled Ibuprofen and Its Interaction with Lipid Membranes

**DOI:** 10.3390/molecules27134127

**Published:** 2022-06-27

**Authors:** Denis S. Baranov, Anna S. Smorygina, Sergei A. Dzuba

**Affiliations:** 1V.V. Voevodsky Institute of Chemical Kinetics and Combustion, SB RAS, 630090 Novosibirsk, Russia; anna.smor.mr@gmail.com; 2Department of Physics, Novosibirsk State University, 630090 Novosibirsk, Russia

**Keywords:** non-steroidal anti-inflammatory drug, NSAID, lipid bilayer, EPR, ^2^H ESEEM

## Abstract

Ibuprofen is a non-steroidal anti-inflammatory drug possessing analgesic and antipyretic activity. Electron paramagnetic resonance (EPR) spectroscopy could be applied to study its interaction with biological membranes and proteins if its spin-labeled analogs were synthesized. Here, a simple sequence of ibuprofen transformations—nitration, esterification, reduction, Sandmeyer reaction, Sonogashira cross-coupling, oxidation and saponification—was developed to attain this goal. The synthesis resulted in spin-labeled ibuprofen (ibuprofen-SL) in which the spin label TEMPOL is attached to the benzene ring. EPR spectra confirmed interaction of ibuprofen-SL with 1-palmitoyl-2-oleoyl-sn-glycero-3-phosphocholine (POPC) bilayers. Using ^2^H electron spin echo envelope modulation (ESEEM) spectroscopy, ibuprofen-SL was found to be embedded into the hydrophobic bilayer interior.

## 1. Introduction

Ibuprofen, along with aspirin and paracetamol, is the most significant non-steroidal anti-inflammatory drug, and is widely used for analgesic and antipyretic purposes. The over-the-counter status of these drugs are based on years of extensive research into their efficacy and safety [1,2,3]. In particular, the general pattern of blocking prostaglandin synthesis via inhibition of membrane enzyme cyclo-oxygenase (COX) by ibuprofen, as well as the parallel side effects caused by this inhibition, mainly in the gastrointestinal tract, has been studied in detail [4]. The other beneficial properties of ibuprofen, such as slowing down neurodegenerative diseases [5,6,7] and cancer [8,9,10,11] have also been shown.

Improvement of ibuprofen-containing drugs is carried out in several directions: increasing its bioavailability [12,13], its use in combination with other drugs [14,15], reducing the negative impact on the gastrointestinal tract [16], and others. In this regard, new data on the processes of interaction of ibuprofen with lipid membranes and with proteins, previously inaccessible for detection and description, could be extremely useful. Different methods, such as X-ray diffraction/scattering [17,18], differential scanning calorimetry [19], quasielastic neutron scattering [20], fluorescence [21], neutron spin echo [22], electrochemical impedance [23], Raman microspectroscopy [24], vibrational sum frequency spectroscopy [25], powder NMR [26] and electron paramagnetic resonance (EPR) spectroscopy [27,28] are used to study ibuprofen-mediated changes in model lipid membranes.

Spin-label EPR spectroscopy [29] allows one to obtain structural and dynamical information at the molecular level, of the kind that is accessible only with this technique. In particular, a pulsed EPR version of electron spin echo envelope modulation (ESEEM) spectroscopy probes the position of molecules in the membranes [30,31,32,33], while double electron–electron resonance (DEER, also known as PELDOR) spectroscopy probes molecular self-assembling in membranes and other biological systems [34,35,36,37,38].

For EPR applications, employment of spin-labeled ibuprofen would be desirable. Usually the nitroxyl label is introduced into ibuprofen through the transformation of the carboxyl group: condensation reactions of ibuprofen with amino or hydroxyl derivatives of nitroxides under the action of dehydrating agents (1-ethyl-3-(3-dimethylaminopropyl)carbodiimide or N,N′-dicyclohexylcarbodiimide/N,N-dimethyl-4-aminopyridine) were used to synthesize esters [39] and amides [40], respectively. A significant disadvantage of this approach is the replacement of an important functional group in the molecule, namely the carboxyl group, with an ester or amide group. In such cases, there is a change in the rate and efficiency of drug–target interactions, since the inhibition of COX is achieved, among other ways, through the formation of critical bonds of its active site with the carboxyl group of ibuprofen [41,42]. In addition, the carboxyl group is the hydrophilic “anchor” of ibuprofen, playing a key role in the drug’s interaction with the lipid membranes and its components [43,44].

Therefore, introduction of a substituent in the benzene ring to preserve basic properties of the ibuprofen (the most important is amphiphilicity of the molecule) would be reasonable. Here, we report a six-step synthesis of a new ibuprofen spin-labeled derivative (ibuprofen-SL), containing a nitroxyl radical 2,2,6,6-tetramethylpiperidin-1-yl (TEMPOL) in the meta position relative to the 1-carboxyethyl group. Also, interaction of ibuprofen-SL with a model membrane of 1-palmitoyl-2-oleoyl-sn-glycero-3-phosphocholine (POPC) was characterized by conventional and pulsed EPR.

## 2. Results and Discussion

### 2.1. Synthesis

The route of synthesis is presented in Figure 1. We used the Sonogashira cross-coupling of 4-ethynyl-2,2,6,6-tetramethylpiperidin-4-ol with iodo-ibuprofen methyl ester **2** to assemble a precursor **3**, which, after oxidation with hydrogen peroxide followed by saponification with aqueous sodium hydroxide solution in one pot, gave the target product **1**. This order of transformations is a good alternative to the use of ethynyl-nitroxides in cross-coupling, which is usually difficult due to the side processes resulting in low yields of spin-labeled products [45,46,47]. Standard procedures for the oxidative iodination (I_2_/HIO_3_/AcOH) of ibuprofen have resulted in an inseparable mixture of 2- and 3-iodo-ibuprofen (Appendix A). Therefore, the necessary compound **2** was obtained in several steps. Initially, ibuprofen was carefully nitrated with nitric acid in sulfuric acid at −5 °C in 89% yield. Esterification of the nitro compound **4** with methanol in the presence of sulfuric acid gave the ester **5** (98% yield). The amine **6** was obtained in 92% yield by reduction with iron filings in aqueous methanol at 65°C. In the next step, the amine **6** was diazotized with sodium nitrite in hydrochloric acid and subjected to Sandmeyer reaction with potassium iodide in water, to prepare the iodo-ibuprofen methyl ester **2** (51% yield).

The structure and purity of the obtained compounds **1**–**6** were confirmed by elemental analysis, IR, ^1^H-NMR, ^13^C-NMR and mass spectrometry (see Appendix A).

### 2.2. Conventional EPR Spectra: Interaction with the POPC Membrane

EPR spectrum of ibuprofen-SL dissolved in toluene at concentration of 1 mM was recorded at room temperature (25 °C)—see Figure 1. This spectrum is typical for the nitroxide rapidly tumbling in a solution. The hyperfine interaction *a* constant directly obtained from the spectrum is 1.5 ± 0.02 mT.

Figure 1 also shows room-temperature EPR spectra for ibuprofen-SL in the presence of POPC bilayers, for the samples obtained in two ways. First, bilayers were prepared from an initial ibuprofen-SL/POPC mixture (1:99 mol/mol). Secondly, a dimethyl sulfoxide (DMSO) solution of ibuprofen-SL was added to pure POPC bilayers. One can see from Figure 1 that both methods of sample preparation result in very similar EPR spectra. A slight difference between them is the presence of a small admixture of a narrow triplet for the second kind of the sample. This narrow triplet obviously belongs to ibuprofen-SL which did not enter the membrane but remains in the water phase. Also, a slight narrowing of the EPR lines is here seen; as DMSO is less viscous than the lipid bilayer, this narrowing can be readily explained by the plasticization effect of DMSO.

The EPR spectra in the POPC bilayer show that molecular motion is hampered remarkably, as compared with the freely rotating case of toluene solution (see Figure 1). This retardation is typical for EPR spectra of spin-labeled molecules in room-temperature POPC membrane—see, e.g., [48]. Therefore, we may conclude that in our samples the ibuprofen-SL molecules interact with the membranes.

### 2.3. Pulsed EPR: Location in the POPC Membrane

Information about location of the ibuprofen-SL molecule in the membrane can be obtained using pulsed EPR. The approach used [30,31,32,33] employs ESEEM spectroscopy. The ESEEM effect appears because of an anisotropic hyperfine interaction of the unpaired electron of the spin label with the nearby nuclei; its magnitude is determined by the number of nuclei located in the immediate proximity of the spin label—at distances less than 1 nm [30,31,32,33]. ^2^H ESEEM of D_2_O-hydrated membranes allows one to determine the location of the spin label respectively to the membrane surface.

The ESEEM signal time trace for ibuprofen-SL in D_2_O-hydrated POPC bilayers, obtained as a function of the delay *t* (see Experimental), *E*(*t*), was refined from the influence of the background spin relaxation by normalizing to the mean echo decay, <*E*(*t*)>, [30,31,32,33]:(1)$$E_{n}(t)=\frac{E(t)}{<E(t)>}-1.$$

The result is shown in Figure 2a. Also in Figure 2a are the analogously obtained reference data on stearic acids, doxyl-spin-labeled at the *n*-th carbon atom positions along the carbon chain, *n*-DSA, and the analogous data [49] on 2-oleoyl-1-palmitoyl-*sn*-glycero-3-phospho(TEMPO)choline (denoted as TEMPO-PC), in which the spin label is attached to the polar lipid head. The corresponding Fourier transforms (see [32]) are given in Figure 2b.

One can see from the data presented in Figure 2 that ESEEM amplitudes for all the samples are different. The largest amplitude is for the TEMPO-PC sample which can be readily understood because the spin label here is directly exposed to the water shell. For the spin-labeled stearic acids, *n*-DSA, the signal amplitude becomes subsequently smaller with the *n* increase, which corresponds to increasing the distance from the membrane surface.

The ESEEM amplitude for ibuprofen-SL in Figure 2 demonstrates its closeness to that of the 5-DSA sample. It is known from molecular dynamics (MD) simulations [50], that the carbon atom positions in the spin-labeled stearic acids resemble approximately those for the host bilayer lipid (with some minor deviation). Note that in the 5-DSA molecule the location of the nitroxide is fixed by its covalent attachment to the alkane chain, whereas in ibuprofen-SL there could be a broader distribution including molecules buried deep into the membrane and others near the surface. Nevertheless, this closeness allows one to state that the spin label in the ibuprofen-SL molecule is located mostly in the membrane interior.

This statement is to be compared with the results of MD simulations for ibuprofen embedded in lipid bilayers [28,51,52] which indicate that ibuprofen molecules prefer to be located in the hydrophobic interior. This is not surprising because, although a hydrophilic carboxylate group of ibuprofen interacts with lipid hydrophilic heads, its large non-polar residue is indeed expected to be located in the hydrophobic interior.

Therefore, the presence of a spin label does not remarkably change the hydrophobicity of the backbone of the parent ibuprofen molecule, which might lead to its excursion from the membrane interior. Therefore ibuprofen-SL molecules can adequately mimic the parent ibuprofen molecules embedded into the biological membrane.

## 3. Materials and Methods

### 3.1. General Information

Elemental analysis was performed with a CHN-analyzer (Model 1106, “Carlo Erba”, Italy). NMR spectra were recorded on a Bruker AV-400 (400 (^1^H) and 101 (^13^С) MHz) or AV-500 (500 (^1^H) and 126 (^13^С) MHz) spectrometer in the CDCl_3_. Chemical shifts (*δ*) are given in ppm with reference to the residual signals of CDCl_3_ (*δ* = 7.26 ppm for ^1^H and *δ* = 77.16 for ^13^C ppm). Melting points were determined with a 1101D Mel-Temp Digital Melting Point apparatus (Electrothermal). HRMS were measured on a Thermo Electron Corporation DFS mass spectrometer (70 eV), using direct injection, the temperature of the ionization chamber was 220–270 °C. The IR spectra were recorded on a Shimadzu IRTracer-100 instrument with GS10802-X Quest ATR ZnSe Accessory (Specac). Column chromatography was performed on silica gel (0.063–0.200 mm). Merck 60 WF254 plates were used for TLC analysis. All reagents and solvents were obtained from commercial sources and used directly after purchased without special purification. Racemic ibuprofen was purchased from Sigma-Aldrich.

### 3.2. Synthesis and Characterization

*4-Ethynyl-2,2,6,6-tetramethylpiperidin-4-ol* was synthesized according to the previously reported procedure [53].

*2-[4-(2-Methylpropyl)-3-nitrophenyl]propanoic acid* (**4**). A solution of ibuprofene (5.5 g, 26.7 mmol) in 100 mL concentrated H_2_SO_4_ was cooled to −5 °C, and mixture of concentrated HNO_3_ (2.15 mL, 32.0 mmol, 67%) and concentrated H_2_SO_4_ (17 mL) was added dropwise for 10 min. The resulting solution was stirred for an additional 20 min. The mixture was added to ice (600 g), and the product was extracted with ethyl acetate (250 mL). The organic layer was separated and dried over MgSO_4_. The solvent was evaporated, and the product was crystallized from hexane. Yield 5.95 g (89%), white solid, mp 91–92 °C. ^1^H-NMR (400 MHz, CDCl_3_) *δ*: 7.83 (d, 1H, H_Ar_, *J* = 1.7 Hz). 7.46 (dd, 1H, H_Ar_, *J* = 7.9 and 1.7 Hz), 7.26 (d, 1H, H_Ar_, *J* = 8.1 Hz), 3.79 (q, 1H, CH, *J* = 7.2 and 7.1 Hz), 2.76 (d, 2H, CH_2_, *J* = 7.1 Hz), 1.89 (p, 1H, CH, *J* = 6.8 and 6.6 Hz), 1.55 (d, 3H, CH_3_, *J* = 7.2 Hz), 0.91 (d, 6H, 2CH_3_, *J* = 6.6 Hz). ^13^С-NMR (126 MHz, CDCl_3_) *δ*: 180.11 (C=O), 149.80 (C_Ar_), 138.84 (C_Ar_), 135.68 (C_Ar_), 133.16 (C_Ar_H), 131.84 (C_Ar_H), 123.93 (C_Ar_H), 44.65, 41.54, 29.57, 22.57, 18.04 (*i*-Bu, CH_3_, CH). IR (film) cm^−1^: 2957, 2870 (Alk), 1701 (C=O), 1524, 1350 (NO_2_). Anal. Calcd. for C_13_H_17_NO_4_: C, 62.48; H, 6.89; N, 5.66. Found: C, 62.14; H, 6.82; N, 5.57.

*Methyl 2-[4-(2-methylpropyl)-3-nitrophenyl]propanoate* (**5**). A solution of compound **4** (5.0 g, 19.9 mmol) and concentrated H_2_SO_4_ (1.5 mL) in methanol (70 mL) was stirred at 40 °C for 20 h. The reaction mixture was poured onto water (400 mL) and petroleum ether (300 mL), the organic layer was separated and dried over MgSO_4_. The solvent was evaporated in vacuo. The crude product was purified by column chromatography on silica gel (eluent: petroleum ether/ethyl acetate, 9:1). Yield 5.18 g (98%), colorless oil. ^1^H-NMR (400 MHz, CDCl_3_) *δ*: 7.79 (d, 1H, H_Ar_, *J* = 1.7 Hz), 7.44 (dd, 1H, H_Ar_, *J* = 7.9 and 1.7 Hz), 7.24 (d, 1H, H_Ar_, *J* = 8.1 Hz), 3.77 (q, 1H, CH, *J* = 7.2 and 7.1 Hz), 3.67 (s, 3H, OCH_3_), 2.74 (d, 2H, CH2, *J* = 7.1 Hz), 1.88 (p, 1H, CH, *J* = 6.7 and 6.7 Hz), 1.51 (d, 3H, CH_3_, *J* = 7.2 Hz), 0.89 (d, 6H, 2CH_3_, *J* = 6.7 Hz). ^13^С-NMR (100 MHz, CDCl_3_) *δ*: 174.12 (C=O), 149.80 (C_Ar_), 139.71 (C_Ar_), 135.27 (C_Ar_), 133.04 (C_Ar_H), 131.61 (C_Ar_H), 123.80 (C_Ar_H), 52.40 (OCH_3_), 44.66, 41.48, 29.53, 22.52, 18.42, (*i*-Bu, CH_3_, CH). IR (film) cm^−1^: 2957, 2870 (Alk), 1736 (C=O), 1529, 1348 (NO_2_), 1207, 1192, 1167 (COC). Anal. Calcd. for C_14_H_19_NO_4_: C, 63.38; H, 7.22; N, 5.28. Found: C, 63.00; H, 7.09; N, 5.19.

*Methyl 2-[3-amino-4-(2-methylpropyl)phenyl]propanoate* (**6**). A mixture of compound **5** (5.0 g, 18.7 mmol), iron filings (4.4 g, 78.6 mmol), NH_4_Cl (2.8 g, 52.3 mmol) and H_2_O (3.8 mL) in methanol (38 mL) was stirred at 65 °C for 20 h. The reaction mixture was poured onto ethyl acetate (350 mL), washed with water (350 mL) and dried with MgSO_4_. The solvent was evaporated in vacuo. The crude product was purified by column chromatography on silica gel (eluent: petroleum ether/ethyl acetate, 9:1). Yield 4.12 g (92%), colorless oil. ^1^H-NMR (400 MHz, CDCl_3_) *δ*: 6.94 (d, 1H, H_Ar_, *J* = 7.7 Hz), 6.64 (dd, 1H, H_Ar_, *J* = 7.7 and 1.6 Hz), 6.61 (d, 1H, H_Ar_, *J* = 1.6 Hz), 3.65 (s, 3H, OCH_3_), 3.61 (q, 1H, CH, *J* = 7.2 and 7.1 Hz), 3.57 (br.s, 2H, NH_2_), 2.34 (d, 2H, CH_2_, *J* = 7.1 Hz), 1.91 (p, 1H, CH, *J* = 6.7 and 6.7 Hz), 1.46 (d, 3H, CH_3_, *J* = 7.2 Hz), 0.95 (d, 6H, 2CH_3_, *J* = 6.6 Hz). ^13^С-NMR (126 MHz, CDCl_3_) *δ*: 175.36 (C=O), 144.53 (C_Ar_), 139.26 (C_Ar_), 131.06 (C_Ar_H), 124.86 (C_Ar_), 117.63 (C_Ar_H), 114.50 (C_Ar_H), 52.08 (OCH_3_), 45.08, 40.66, 27.80, 22.83, 18.67 (*i*-Bu, CH_3_, CH). IR (film) cm^−1^: 3448, 3371 (NH_2_), 2951, 2866 (Alk), 1728 (C=O), 1194, 1161 (COC). HRMS (ESI) *m*/*z*: [M]^+^ Calcd for C_14_H_21_NO_2_ 235.1567. Found 235.1571.

*Methyl 2-[3-iodo-4-(2-methylpropyl)phenyl]propanoate* (**2**). A solution of NaNO_2_ (610 mg, 8.8 mmol) in H_2_O (4 mL) was added dropwise to amine **5** in 8 M HCl (12 mL) at −5 °С. The reaction mixture was stirred for 15 min and added in portions over 10 min to a stirring mixture of KI (1.66 g, 10.0 mmol) in H_2_O (40 mL). Then, the reaction mixture was stirred at 60 °C for 2 h. Water (100 mL) and petroleum ether (200 mL) were added, and the organic layer was separated and dried over MgSO_4_. The crude product was purified by column chromatography on silica gel (eluent: petroleum ether/toluene, 1:1). Yield 1.5 g (51%), colorless oil. ^1^H-NMR (500 MHz, CDCl_3_) *δ*: 7.74 (d, 1H, H_Ar_, *J* = 1.5 Hz), 7.19 (dd, 1H, H_Ar_, *J* = 7.8 and 1.5 Hz), 7.09 (d, 1H, H_Ar_, *J* = 7.8 Hz), 3.67 (s, 3H, OCH_3_), 3.63 (q, 1H, CH, *J* = 7.2 and 7.0 Hz), 2.55 (d, 2H, CH_2_, *J* = 7.2 Hz), 1.95 (p, 1H, CH, *J* = 6.9 and 6.7 Hz), 1.46 (d, 3H, CH_3_, *J* = 7.2 Hz), 0.94 (d, 6H, 2CH_3_, *J* = 6.5 Гц). ^13^С-NMR (126 MHz, CDCl_3_) *δ*: 174.73 (C=O), 143.23 (C_Ar_), 139.95 (C_Ar_), 138.50 (C_Ar_H), 130.49 (C_Ar_H), 127.13 (C_Ar_H), 101.40 (C_Ar_I), 52.30 (OCH_3_), 49.24, 44.46, 28.98, 22.37, 18.63 (*i*-Bu, CH_3_, CH). IR (film) cm^−1^: 2953, 2868 (Alk), 1736 (C=O), 1203, 1163 (COC). HRMS (ESI) *m*/*z*: [M]^+^ Calcd for C_14_H_19_IO_2_ 346.0424. Found 346.0421.

*Methyl 2-{3-[(4-hydroxy-2,2,6,6-tetramethylpiperidin-4-yl)ethynyl]-4-(2-methylpropyl)phenyl}propanoate* (**3**). 4-Ethynyl-2,2,6,6-tetramethylpiperidin-4-ol (290 mg, 1.6 mmol) was added to a solution of iodide 2 (500 mg, 1.4 mmol), PdCl_2_(PPh_3_)_2_ (30 mg, 0.04 mmol), CuI (20 mg, 0.1 mmol), and Et_3_N (2 mL) in toluene (20 mL) and stirred under an argon atmosphere at 75 °С for 1.5 h. After cooling, the reaction mixture was filtered, filtrate was washed with 6 M aqueous ammonia (50 mL) and dried over MgSO_4_. The crude product was purified by column chromatography on silica gel (eluent: ethyl acetate). Yield 300 mg (54%), white solid, mp 95–96 °C. ^1^H-NMR (400 MHz, CDCl_3_) *δ*: 7.30 (d, 1H, H_Ar_, *J* = 1.7 Hz). 7.16 (dd, 1H, H_Ar_, *J* = 7.9 and 1.9 Hz), 7.09 (d, 1H, HAr, *J* = 7.9 Hz), 3.62–3.68 (m, 4H, OCH_3_+CH), 2.58 (d, 2H, CH_2_, *J* = 7.1 Hz), 2.34 (br.m, 2H, NH+OH), 1.91–2.00 (m, 3H, CH_2_+CH), 1.76 (d, 2H, CH2, *J* = 13.4 Hz), 1.46 (d, 3H, CH_3_, *J* = 7.2 Hz), 1.33 (d, 12H, 4CH_3_, *J* = 11.4 Hz), 0.91 (d, 6H, 2CH_3_, *J* = 6.6 Hz). ^13^С-NMR (126 MHz, CDCl_3_) *δ*: 174.95 (C=O), 142.75 (C_Ar_), 137.99 (C_Ar_), 131.35 (C_Ar_H), 130.12 (C_Ar_H), 127.50 (C_Ar_H), 122.65 (C_Ar_), 83.05, 97.62 (C≡C), 67.32 (COH), 52.24, 51.02, 49.63, 44.85, 43.48, 32.84, 31.65, 29.68, 22.62, 18.59 (*i*-Bu, CH_3_, CH, OCH_3_, NH(C(CH_3_)_2_CH_2_)_2_). IR (film) cm^−1^: 3500 (NH, OH), 2955, 2870 (Alk), 1738 (C=O), 1163 (COC). Anal. Calcd. for C_25_H_37_NO_3_: C, 75.15; H, 9.33; N, 3.51. Found: C, 75.59; H, 9.51; N, 3.43. HRMS (ESI) *m*/*z*: [M]^+^ Calcd for C_25_H_37_NO_3_ 399.2768. Found 399.2767.

*{4-[5-(1-Carboxyethyl)-2-(2-methylpropyl)phenyl]ethynyl-4-hydroxy-2,2,6,6-tetramethylpiperidin-1-yl}oxidanyl* (**1**). A solution of 30% Hydrogen peroxide (0.37 mL) was added to a mixture compound 5 (120 mg, 0.3 mmol), Na_2_WO_4_·2H_2_O (15 mg, 0.045 mmol), EDTA disodium salt (15 mg, 0.045 mmol) and H_2_O (0.37 mL) in methanol (3 mL) at room temperature and stirred for 70 h. The reaction mixture was filtered, the solvent was evaporated in vacuo, 2 M NaOH (3 mL) was added, and the resulting mixture was stirred at 80 °C for 0.5 h. Then, H_2_O (5 mL) was added. The solution was cooled to 0 °C and treated with 1 M HCl to neutral pH. The crude product was extracted with ethyl acetate and purified by column chromatography on silica gel (eluent: petroleum ether/ethyl acetate, 1:1). Yield 60 mg (50%), red viscous oil. ^1^H-NMR (400 MHz, CDCl_3_) *δ*: 7.10–7.75 (m, 3H, H_Ar_). 3.78 (br.m, 1H, CH), 2.70 (br.m, 2H, CH_2_), 2.08 (br.m, 1H, CH), 1.58 (br.m, 3H, CH_3_), 1.02 (br.m, 6H, 2CH_3_). ^13^С-NMR (126 MHz, CDCl_3_) *δ*: 177.63 (C=O), 141.99 (C_Ar_), 136.28 (C_Ar_H), 130.94 (C_Ar_), 128.80 (C_Ar_H), 126.92 (C_Ar_H), 120.36 (C_Ar_), 84.74 44.56, (C≡C), 42.44, 28.64, 28.42, 22.03, 17.14 (*i*-Bu, CH_3_, CH). IR (film) cm^−1^: 3470 (OH), 2931, 2868 (Alk), 2241 (C≡C), 1709 (C=O). HRMS (ESI) *m*/*z*: [M]^+^ Calcd for C_24_H_34_NO_4_ 400.2482. Found 400.2486.

### 3.3. Sample Preparations for EPR Investigation

To obtain EPR spectra in toluene solution, ibuprofen-SL was dissolved in toluene (Ekros-Analytica, St. Petersburg, Russia, distilled before the use) at concentration of 1 mM.

Lipids 1-palmitoyl-2-oleoyl-sn-glycero-3-phosphocholine (POPC) were from Avanti Polar Lipids (Birmingham, AL, USA). Spin-labeled stearic acids 5-DSA, 12-DSA and 16-DSA were from Sigma-Aldrich (Saint Louis, MO, USA). These substances were used as received.

The bilayer samples with ibuprofen-SL were prepared in two ways. For the first method, POPC and ibuprofen-SL were dissolved separately in chloroform, then the two solutions were mixed, so that the ibuprofen-SL/POPC molar ratio in the mixture was 1:99. Then the solvent was removed in a nitrogen stream and the mixture was stored under vacuum for 4 h. Then phosphate-buffered saline (pH 7.0) was added to the resulting powder in a proportion of 10:1 (*v*/*v*), the sample was stirred and then stored for 2 h. Upon this procedure, multilamellar vesicles (MLVs) are formed [54]. Then the mixture was centrifuged to remove excess solvent. For some experiments, deuterium-substituted water has been used instead of ordinary water. For the second method, MLVs were similarly prepared, but without ibuprofen-SL; then, a dimethylsulfoxide (DMSO) solution of ibuprofen-SL was subsequently added. DMSO content was less than 10 vol% respectively to the sample volume; the total ibuprofen-SL quantity was 100 times smaller than the total lipid quantity (mol/mol).

For comparative purposes, POPC bilayers containing 1 mol% of doxyl-spin-labeled stearic acids *n*-DSA (*n* = 5, 12 or 16) were also prepared, in a manner similar to the first method described above.

The prepared samples were put into EPR glass tubes of 3 mm o.d. and studied either at room temperature or at 80 K. In the latter case, the samples were quickly frozen by immersion into liquid nitrogen.

### 3.4. EPR Measurements

Conventional EPR spectra were obtained at room temperature with an X-band EPR benchtop SPINSCAN-X spectrometer (ADANI, Minsk, Belorussia), operating at modulation amplitude of 0.01 mT, with the output microwave (MW) power of 100 mW, the MW attenuation set to −25 dB, and with sweep and constant times of 60 s and 46 ms, respectively.

In pulsed EPR studies, an X-band Bruker ELEXSYS E580 EPR spectrometer was used equipped with a split-ring Bruker ER 4118 X-MS-3 resonator and an Oxford Instruments CF-935 cryostat. A three-pulse ESEEM sequence (*π*/2)-*τ*-(*π*/2)-*t*-(*π*/2)-*τ*-*echo* was employed, with excitation at the maximum of the echo-detected EPR spectrum. The pulse lengths were 16 ns, time delay *τ* was 204 ns, the time delay *t* was scanned from 300 ns to 10 µs, with a 12 or 16 ns time step. The resonator was cooled with a stream of cold nitrogen gas. The temperature was controlled by a nitrogen flow stabilized by a Bruker ER4131VT temperature controller, the sample temperature was kept near 80 K.

## 4. Conclusions

It is shown in this work that ibuprofen—a non-steroidal anti-inflammatory drug possessing analgesic and antipyretic activity—can be labeled by a TEMPOL spin label via a simple six-step sequence of nitration, esterification, reduction, Sandmeyer reaction, Sonogashira cross-coupling, oxidation and saponification. The TEMPOL spin label in ibuprofen is coupled to the benzene core of the molecule via a triple bond.

EPR data show that spin-labeled ibuprofen interacts with lipid membranes. The immersion depth into the membrane was found with pulsed EPR (^2^H ESEEM spectroscopy) in D_2_O-hydrated bilayers, the results indicate that spin label is buried into the membrane interior, which is in agreement with literature MD data on the immersion of the ibuprofen molecule. So, the proposed method of synthesis allows one to obtain spin-labeled ibuprofen that can adequately mimic the parent ibuprofen molecules interacting with the biological membrane.

## Data Availability

Data is contained within the article.

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
