# Peer review of "Synthesis of Spin-Labeled Ibuprofen and Its Interaction with Lipid Membranes"

_molecules, 2022, doi:10.3390/molecules27134127_

Round 1

Reviewer 1 Report

Comments for Authors:

This manuscript by Baranov, Dzuba, and co-workers describes the synthesis of spin-labeled ibuprofen by introducing a TEMPO radical at the meta position of the benzene ring of ibuprofen to study its interaction with lipid membranes using EPR. Since ibuprofen is one of the most widely used analgesics and antipyretic drugs, the study of the interaction of ibuprofen with lipid membranes holds great importance and promise for improving its performance and getting more insights into its metabolic path. Although the spin-labeled ibuprofen derivatives have been reported and used as agents for EPR study, the main novelty of this study is that an alternative ibuprofen analog containing a nitroxyl radical (TEMOP·) was prepared and used to study its interaction with biological membranes via EPR, which could largely prevent the change of the original drug-targeting anchor (the carboxylic acid group) during the introduction of spin-labeling. Thus, the novelty of this work seems to merit the standard and this manuscript will surely be of interest to a wide audience in the ibuprofen-related research. However, the scientific significance and advantages of using spin-labeled ibuprofen via EPR to study the interaction with lipid membranes and proteins are unclear. The manuscript could not be published in its current form unless several key issues are fully resolved by the authors. The concerns from this reviewer have been listed.

-        The introduction section, page 1 lines 36-41. The authors state that X-ray diffraction/scattering, differential scanning calorimetry, quasielastic neutron scattering, fluorescence, etc. have been used for the study of ibuprofen-mediated changes in model lipid membranes. The authors strongly suggested further clarifying the superiority of electron paramagnetic resonance (EPR) spectroscopy compared with the other mentioned technical approaches for the same purpose. This would greatly help for clarifying the significance of developing spin-labeled ibuprofen in this work. 

-        The key novelty of this work is the development of an alternative ibuprofen analog containing a nitroxyl radical (TEMOP·) at the meta-position of the benzene ring on ibuprofen rather than the transformation of the carboxyl group, which is the hydrophilic anchor” of ibuprofen for protein or lipid membranes binding. However, the introduction of nitroxyl radical (TEMOP·) via post-modification also largely change the hydrophobicity of the backbone of ibuprofen, which also may change the interactions of ibuprofen inside the hydrophobic interior. If that is the case, the insight obtained from this spin-labeled ibuprofen is also meaningless. At least, the results obtained by using this newly synthesized spin-labeled ibuprofen could be compared with the results by using previous spin-labeled ibuprofen. 

-        Page 4, lines 141-148, the conclusion “ibuprofen molecules prefer to be located in the hydrophobic interior” was only supported by the comparison of ESEEM and molecular dynamics (MD) simulations from other reported literatures. This is difficult to convince this reviewer. This newly synthesized spin-labeled ibuprofen also has a negatively charged hydrophilic carboxylate group, which may interact with lipids' hydrophilic heads.

-        Page 2, lines 75 and 83, Supplementary Materials have been mentioned and corresponded with the characterization data for new compounds. However, the Supplementary Materials can not be found and downloaded.

Examples of typos and unclear sentences:

-       Page 1, introduction section, line 25. is based on should be “are based on”. 

-       Page 2, introduction section, line 30 “has” should be “have”. 

The Languages, grammar, and other typos should be checked and corrected.

Author Response

Response to Reviewers

Reviewer # 1

Comment.        The introduction section, page 1 lines 36-41. The authors state that X-ray diffraction/scattering, differential scanning calorimetry, quasielastic neutron scattering, fluorescence, etc. have been used for the study of ibuprofen-mediated changes in model lipid membranes. The authors strongly suggested further clarifying the superiority of electron paramagnetic resonance (EPR) spectroscopy compared with the other mentioned technical approaches for the same purpose. This would greatly help for clarifying the significance of developing spin-labeled ibuprofen in this work. 

Answer. We introduced a new paragraph (accompanied with new references) below the mentioned text which describes advantages of EPR technique:

               “Spin-label EPR spectroscopy [29] allows obtaining structural and dynamical information at the molecular level, of the kind that is accessible only for this technique. In particular, its pulsed versions Electron Spin Echo Envelope Modulation (ESEEM) probes position of the molecules in the membranes [30-33], and Double Electron-Electron Resonance (DEER, also known as PELDOR) [34-38] probes molecular self-assembling in membranes and other biological systems. ”

 Comment.       The key novelty of this work is the development of an alternative ibuprofen analog containing a nitroxyl radical (TEMOP·) at the meta-position of the benzene ring on ibuprofen rather than the transformation of the carboxyl group, which is the hydrophilic “anchor” of ibuprofen for protein or lipid membranes binding. However, the introduction of nitroxyl radical (TEMOP·) via post-modification also largely change the hydrophobicity of the backbone of ibuprofen, which also may change the interactions of ibuprofen inside the hydrophobic interior. If that is the case, the insight obtained from this spin-labeled ibuprofen is also meaningless. At least, the results obtained by using this newly synthesized spin-labeled ibuprofen could be compared with the results by using previous spin-labeled ibuprofen. 

Answer. Yes, it could be indeed become a problem. However, our ESEEM data, along with the comparison with literature MD simulations show that possible change of the hydrophobicity of the backbone of ibuprofen is not critical in our case (lines 141-148 of our original submission). To make our reasoning clearer and the statement bolder, we introduce at the end of this text a new paragraph:

“So, the presence of spin label does not change remarkably the hydrophobicity of the backbone of the parent ibuprofen molecule, which could lead to its excursion from the membrane interior, and therefore ibuprofen-SL can adequately mimic the parent molecule embedded into the biological membrane.”

And we have no in our disposal spin-label ibuprofen that is described previously in literature, to perform the similar ESEEM experiments.

Comment.        Page 4, lines 141-148, the conclusion “ibuprofen molecules prefer to be located in the hydrophobic interior” was only supported by the comparison of ESEEM and molecular dynamics (MD) simulations from other reported literatures. This is difficult to convince this reviewer. This newly synthesized spin-labeled ibuprofen also has a negatively charged hydrophilic carboxylate group, which may interact with lipids' hydrophilic heads.

Answer. To better clarify the situation, we added the sentence in the last but one paragraph of the 2.3 Subsection:

“It is not surprising, because although a hydrophilic carboxylate group of ibuprofen interacts with lipids' hydrophilic heads its large non-polar residue certainly is expected to be located in the hydrophobic interior.”

Comment.        Page 2, lines 75 and 83, Supplementary Materials have been mentioned and corresponded with the characterization data for new compounds. However, the Supplementary Materials can not be found and downloaded.

Answer. Sorry, it was our mistake. Now the Supplementary Materials can be downloaded.

Examples of typos and unclear sentences:

-       Page 1, introduction section, line 25. “is based on” should be “are based on”. 

-       Page 2, introduction section, line 30 “has” should be “have”. 

The Languages, grammar, and other typos should be checked and corrected.

Answer. Thanks, we tried to do our best to improve language and grammar.

Reviewer 2 Report

Please find my remarks in the attachment.

Kind regards.

Author Response

Response to Reviewers

Reviewer # 2

Comment 1. There could be a brief discussion of the limitations of the ESEEM experiment for probing the ibuprofen-SL in the membrane/bilayer. It’s an ensemble average method, thereby the similarity of ibuprofen-SL to 5-DSA might be misleading. In 5-DSA, the location of the nitroxide is fixed by its covalent attachment to the alkane chain, whereas in ibuprofen-SL there could be a broad distribution including molecules buried deep into the membrane and other near the surface.

Answer. We agree, and added a sentence reflecting this fact in the last but one paragraph at the end of the 2.3 Subsection:

“Although in 5-DSA the location of the nitroxide is fixed by its covalent attachment to the alkane chain, whereas in ibuprofen-SL there could be a broader distribution including molecules buried deep into the membrane and other near the surface, nevertheless this closeness allows stating that spin label in the ibuprofen-SL molecule is located mostly below the membrane surface.”

Comment 2. The isotropic g factor, practically identical to the free electron g factor can’t be correct. Typically, nitroxides have isotropic g values on the order of 2.005. If you want to keep the g factor in the manuscript, there should be a comparison to a reference molecule with known g factor, leading to a substantially more reliable value.

Answer. It is true, it was an error in the spectrometer software, we removed this value from our manuscript, thank you.

Comment 3. I don’t understand the statement in line 102: “this narrowing can be readily explained by the plasticization effect of the DMSO presence”. Here, a bit more of an explanation would help (potentially also a citation for the plasticization effect of DMSO).

Answer. As an argument, we added in this part of our MS the indication that DMSO is less viscous than the lipid bilayer.

Figures

Figure 1: This figure is a bit confusing. The red, dashed line is easily mistaken for a simulation (one could also consider to include simulation of the cw EPR spectra), especially since it is superimposed on the black line. Unlike for thee other lines, no labelling is given for the red, dashed line. It would be helpful to clearly distinguish and label all lines in the figure.

Answer. OK, we changed this figure.

Figure 2: The green line should be labelled ibuprofen-SL.

Answer. It is corrected.

Typos

Line 109: “retardant” should be replaced by hampered, hindered or something similar (retardant is not an adjective).

Line 137: TEMPO-PC (instead of “TENPO-PC”) Line 140: which corresponds (instead of “which is corresponds)”

Line 274: before use (instead of “before the use”)

Line 280: in a nitrogen stream

Line 283: excess solvent

Answer. It is corrected.

Round 2

Reviewer 1 Report

I have carefully read the revised manuscript and the responses provided by the authors. They have addressed the comments of the referees with sufficient data analysis, and the significance of this work has also been well-organized. After the English language and style have been further checked and improved, I think that this work is qualified to be published. 

I just provide an additional minor remark for Supplementary Materials. In the main text (page 3, lines 98-99), mass spectrometry data were mentioned. However, there is no mass spectrometry of the compound provided in the attached Supplementary Materials.

Author Response

We tried to improve our language.

We added mass spectrometry data to th Suplemantary Materials.
